# Explainable Machine Learning Predictions for the Long-term Performance of Brain Computer Interfaces

## Abstract

Brain computer interfaces (BCIs) can decode neural signals to control assistive technologies such as robotic limbs for people with paralysis. Neural recordings from intracortical microelectrodes offer the spatiotemporal resolution (e.g., sortable units) necessary for complex tasks, such as controlling a robotic arm with multiple degrees of freedom. However, the quality of these signals decays over time despite many attempts to prolong their longevity. This decrease in long-term performance limits the implementation of this potentially beneficial technology. Predicting whether a channel will have sortable units across time would mitigate this issue and increase the utility of these devices by reducing uncertainty, yet to our knowledge, no such methods exist. Similarly, it would be useful to understand how variables like time post-implantation, electrochemical characteristics, and electrode design impact the long-term quality of these signals. Here, we obtained longitudinal neural recordings and electrochemical data from freely behaving rats implanted with a custom designed microelectrode array with varying site areas, shank positions, and site depths. This dataset was used to develop an explainable machine learning pipeline that predicts with high accuracy the presence of sortable units on a given channel pre-recordings and elucidates the most important factors leading to these predictions. Our pipeline was able to predict whether a channel will be active with an AUC of 0.79 (95% C.I. 0.73–0.86) on unseen data. The most important features of the model were experimental subject, time post-implantation, and a channel's previous spike metrics. Electrode site depth was the most important electrode design variable. Our results demonstrate the feasibility of implementing explainable machine learning pipelines for longitudinal BCI studies and support previous reports on how factors like time, inter-animal variability, and cortical depth impact long-term performance of BCIs. These results are an important step forward in improving efficient decoding performance and guiding device development, which stand to advance the field and benefit the lives of human BCI patients.

## 1 Introduction

Decoding neural signals through brain computer interfaces (BCI) can improve quality of life for people with paralysis. Every year, approximately 20,000 patients suffer from spinal cord injury in the US alone (Sekhon & Fehlings, 2001). Most of these cases result in tetraplegia, causing paralysis from the neck down (Sekhon & Fehlings, 2001). By interfacing directly with the brain, BCIs can be used as assistive technologies for these patients. Neural signals can be decoded and used to move computer cursors (Hochberg et al., 2006), control robotic limbs (Hochberg et al., 2012), and enrich exoskeleton (Benabid et al., 2019) and spinal cord stimulation technologies (Capogrosso et al., 2016) that aim to restore locomotion. Neural signals can be recorded using a variety of interfaces that vary in their degree of invasiveness (Eisinger et al., 2018). Recording signals directly from the brain cortex using implantable microelectrodes enables isolating spikes from individual neurons (single units). This level of precision is important for complex implementations such as controlling a robotic arm with multiple degrees of freedom (Lebedev & Nicolelis, 2017) and has led to the creation of companies like Elon Musk's Neuralink (Musk, 2019).

One challenge for intracortical BCIs is the long-term stability of the neural signals. The ability to record sortable units gradually decreases over time, compromising decoding performance due to reduced quality of the recorded neural signals (Williams et al., 1999; Colachis et al., 2021; Downey et al., 2018). Potential solutions, such as replacing the electrode, might require a second surgery, hindering the feasibility and clinical implementation of BCI technologies. There have also been attempts to mitigate this decline in stability by using drug delivery systems, electrode coatings, and new electrode materials (Colachis et al., 2021), but the problem persists. To date, we still do not fully understand how these factors, and others such as the electrode design, inter-animal variability, or electrochemical features, might play a role in the stability of these signals. Predicting when stability of neural signals will decrease and understanding which features of the interface contribute to this, would benefit clinicians and researchers and improve utility of intracortical BCIs.

Classical statistical techniques that focus on the individual impact of each of these factors may be ineffective due to the size and complexity of many BCI datasets. These rich datasets may benefit from more advanced machine learning (ML) techniques. In recent years, ML has been used used in medicine for the prediction of disease onset (Fleuren et al., 2020; Yahaya et al., 2020) and for the analysis of neurotherapeutics such as Deep Brain Stimulation (DBS) for Parkinson's Disease (Peralta et al., 2021). However, to our knowledge, no ML studies have been applied on the longitudinal stability of intracortical BCIs. In addition to predicting when a recording channel will be active (has sortable units), applying ML explainability tools could also shed light on the most important factors involved in these predictions, such as time post-implantation, electrochemical characteristics, and electrode design variables.

Here, we obtained longitudinal neural recordings and electrochemical data (voltage transient, impedance spectroscopy) from freely behaving rats implanted with a custom made intracortical microelectrode (Fig. 1). This device has 16 channels with varying shanks, electrode-site areas, and cortical depths. We only selected features that were available prior to a given recording session and that are commonly available in clinical BCI studies. Following offline spike sorting, we developed an explainable ML pipeline that predicts (pre-recordings) whether a channel will be active and elucidate the most important features leading to these predictions (Fig. 1). We bypassed the complexity/interpretability tradeoff of many black box ML models by using gradient boosting (Friedman, 2001). This non-parametric model offers high accuracy on tabular datasets (Shwartz-Ziv & Armon, 2022) and is still interpretable (Lundberg et al., 2018). Through the use of Shapley additive explanations (SHAP), a method derived from game theory (Lundberg & Lee, 2017), we obtained local and global explanations that shed light on the most important factors for individual predictions, and the model as a whole, respectively. Similarly, through the use of decision paths, we elucidated how feature importance changes depending on the presence or absence of sortable units in chronic timepoints. Overall, these findings demonstrate the feasibility of predictive modeling and ML explainability tools for the longitudinal performance of BCIs and similar neurotechnologies. Implementation of these tools can guide the design of future BCI studies (including non-intracortical BCI experiments) as well as the manufacturing of novel intracortical microelectrodes. Understanding the most important factors affecting the long-term stability of BCIs and predicting when a channel is active has the potential to propel forward the translatability of these assistive technologies for millions of patients with paralysis.

## 2 METHODS

### 2.1 OVERVIEW

Following the collection of intracortical recordings and electrochemical characteristics over a period of 15 weeks with a custom-made microelectrode (Fig. 1), we developed an explainable ML pipeline that relied on gradient boosting and Shapley additive explanations. Importantly, all input features of the model are available prior to the recording session (Fig. 1) and are routinely collected in clinical BCI studies (i.e. impedance). This pipeline not only predicted with high accuracy whether a channel would have sortable units, it also elucidated the most important features leading to such prediction.

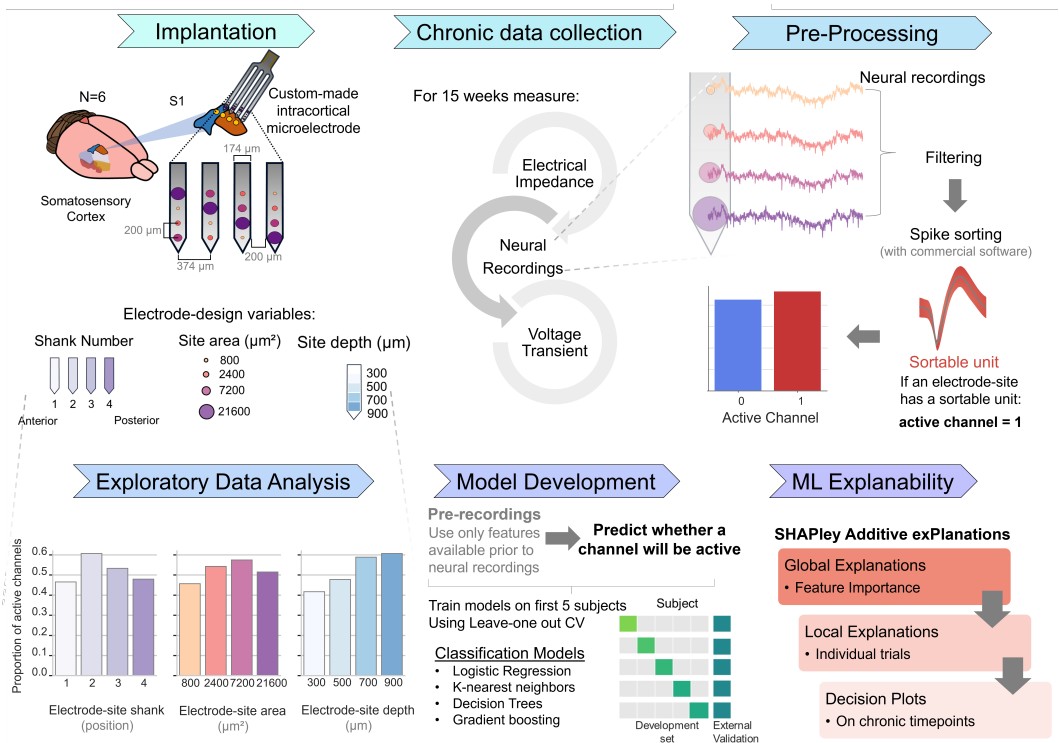

Figure 1: Data collection and Explainable Machine Learning Pipeline. Implantation: A silicon microelectrode with a custom design was implanted in the S1 of rats. The electrode-sites of this device are combination of are shank, area, and depth. Chronic data collection: electrochemical features and neural recordings were obtained for up to 15 weeks post-implantation. Pre-processing: following offline sorting, channels that had sortable units were deemed active. EDA: Percentage of active sites across electrode design variables. Model development: in order to predict whether a channel will be active prior to the recording session, only features available prior to the collection of neural recordings were selected, such as the channel's previous sortable unit metrics. The development set used to train each classifier was done using leave-one out cross-validation and performance was evaluated on an external validation subject. ML Explainability: SHAP values were computed to determine the most important predictors for the model, on chronic timepoints, and for individual trials.

## 2.2 SURGICAL IMPLANTATION AND CHRONIC MEASUREMENTS

A custom silicon microelectrode array with variable site sizes distributed across 4 shanks was designed for this study (NeuroNexus, Ann Arbor, MI). Following sterilization with ethylene oxide, this device inserted into the somatosensory cortex of adult Sprague Dawley rats using a micro-insertion system, resulting in a maximum electrode site depth of 900 μm (see: Fig. 1). All surgeries were carried out by the same surgeon. The first 5 subjects were implanted within a period of 2 weeks. The sixth animal was implanted a month afterwards and was used as the external validation set in our machine learning pipeline (Fig. 1). Specifics of the surgical procedure are explained elsewhere (Urdaneta et al., 2021).

Following implantation, neural recordings, electrical impedance spectroscopy (EIS), and voltage transients (VT) were measured for 15 weeks (Fig. 1). First, neural recordings from all 16 channels were taken for 5 minutes at a sampling rate of 24 kHz on freely behaving animals using a PZ5 amplifier (Tucker Davis Technologies, Alachua, FL). Immediately after, EIS 15 mV peak-to-peak measurements were obtained using logarithmic frequency sweeps from 10 Hz to 100 kHz using a PG-STAT-128N Potentiostat (Metrohm, Utrecht, Netherlands). Lastly, VT was measured using 5 μA symmetric waveforms with a phase duration of 50 μs using a IZ-32 stimulator (Tucker-Davis Technologies, Alachua, FL) (Saldanha et al., 2021).

## 2.3 VARIABLES AND FEATURE ENGINEERING

To determine whether a channel was active (presence of sortable units), raw neural signals were sorted for spike waveforms using Offline Sorter (Plexon Inc. Dallas, TX) (Fig. 1). This supervised method was chosen over automatic alternatives since it allows for easy artifact removal (typical of recordings in freely behaving animals) and has been widely used in the field (O'Doherty et al., 2009). Only channels with spike waveforms that passed the inspection of two blind reviewers and had more than 200 spikes, were considered active. Importantly, our threshold for classifying a channel as active was solely dependent on the presence of a sortable waveform. In other words, having a sortable unit in a channel does not imply that the extracellular recordings are coming from a single, individual neuron. Spike metrics (SNR, amplitude, and spike count) were calculated automatically with the Offline Sorter software (Plexon Inc. Dallas, TX). Overall, 52.1% of all channels across the duration of the study were active. Several features were engineered to potentially augment the model's predictive power. First, we obtained the electrochemical (EIS and VT) and spike metrics of the first recording session that had a sortable unit for a particular channel ("First S.U."). These spike metrics were amplitude, signal-to-noise ratio (SNR), and spike count. In the case of intracortical microstimulation (ICMS), metrics such as the amplitude of the first detection threshold, serves as a good proxy for the overall chronic performance of that channel (Urdaneta et al., 2022). Therefore, obtaining the metrics of the first time the channel was active might be informative for the model. Similarly, we obtained electrochemical and spike metrics of the previous experimental session. To avoid potential data leakage, features such as "week of previous active channel" were dropped. Electrochemical features (EIS and VT) were imputed using last observation carried forward (LOCF) imputation. Feature scaling was performed for the K-nearest neighbor (KNN) and logistic regression algorithms using min-max normalization (Patro & Sahu, 2015). However, given that decision tree models are insensitive to data variance (Dietterich & Kong, 1995) normalization was not performed for either gradient boosting and decision tree classifier models. Lastly, cross-correlated variables (i.e., day vs. week vs. month post-implantation) were removed, while some highly correlated variables such as different impedance frequencies were left untouched for the explainability pipeline. Fig. 4 and table 2 (appendix) show a correlation matrix with the final list of features, and descriptions of those features, respectively.

## 2.4 MODEL DEVELOPMENT

In order to capture the nonlinear interactions from this type of dataset we used gradient boosting, a non-parametric ensemble model that uses gradient descent in the function space Friedman (2001). We compared the performance of gradient boosting with baseline classifiers. Specifically, we used a KNN classifier with 5 neighbors and uniform weights, a Gini entropy decision tree classifier, and a logistic regression with ridge penalty. All these models were implemented from the Scikit-Learn library Pedregosa et al. (2011). We used XGBoost (extreme gradient boosting), an implementation of gradient boosting known for its improved speed (Chen & Guestrin, 2016) as well as its ability to outperform even complex deep learning models on tabular datasets (Shwartz-Ziv & Armon, 2022). Hyperparameter optimization was performed using Optuna, an automatic search space for tuning hyperparameters based on dynamically constructed spaces (Akiba et al., 2019). The final hyperparameters for our gradient boosting model were: number of estimators: 1190, learning rate 0.003, subsample = 74.8%, and a depth-wise growth policy. To take into account the performance heterogeneity observed across subjects in previous BCI studies (Williams et al., 1999; Downey et al., 2018), and to better generalize on unseen data, we created a development set with the first 5 implanted subjects and an external validation set with the last-implanted animal. All models were trained using leave one out cross-validation (Kearns & Ron, 1997) on the development set, and its performance was compared to the external validation set.

## 2.5 MODEL PERFORMANCE

We used different metrics to evaluate the performance of each classifier across the development and external validation sets. Accuracy and AUC-ROC (area under the receiving-operating characteristics curve) were used as performance metrics. Given that the dataset is slightly imbalanced, 52.1% for the positive (channel active = 1) class. We decided to also use metrics that performed well in imbalanced datasets. For this we used both precision and recall, as well as the F1 score, the harmonic

mean between precision and recall. Furthermore, each dataset was bootstrapped (resample with replacement) 1000 times to obtain 95% confidence intervals for each metric.

## 2.6 EXPLAINABLE MACHINE LEARNING

The last step of our explainable ML pipeline was the use of SHAP (Lundberg & Lee, 2017), a concept derived from game theory (Charnes et al., 1988). This method quantifies how predictions for a particular model are affected after the removal of an individual feature in an iterative fashion (Lundberg & Lee, 2017). This way SHAP assigns a relative importance value for each feature on individual predictions. Each SHAP value $\phi_i$ is given by:

$$\phi_i = \sum_{S \subseteq N \setminus \{i\}} \frac{|S|!(M - |S| - 1)!}{M!} \left[ f_x(S \cup \{i\}) - f_x(S) \right]$$

where $S$ is the set of non-zero indexes, $M$ is the number of input features, and $N$ is the set of all input features (Lundberg & Lee, 2017). Our gradient boosting model was passed through a Tree Explainer (Lundberg et al., 2018) in order to obtain the global importance of the model, individual instances, and decision paths.

## 3 RESULTS

### 3.1 THE PROPORTION OF ACTIVE CHANNELS MONOTONICALLY INCREASES WITH ELECTRODE SITE DEPTH

We calculated the proportion of active channels (channels with sortable units) for each distinct feature (area, shank, depth) of our custom-designed intracortical microelectrode (Fig. 1 - exploratory data analysis). The results of this analysis showed that channel depth had a monotonic increase in the proportion of active channels as a function of cortical depth. Specifically, there were 28.4% more active channels for sites 900 μm from the cortical surface (134) than 300 μm from cortical surface (96). Proportionally, only 41.7% of superficial sites (300 μm) were active compared to 60.6% of deep sites (900 μm). In contrast, the percent difference in the number of active sites between the largest site-area (21,600 μm²) (105) and the smallest site-area (800 μm²) (118) was only 11.01%. The site area with the highest proportion of active sites was 7200 μm², the second largest area, with 57.5%. Lastly, the proportion of active sites for the most anterior and posterior and most shank was similar, with 46.5% and 47.9% for positions 1 and 4, respectively. The shank position with the highest proportion of active channels was the second most anterior (position 2) with 60.6%. Overall, compared to site-size and shank position, electrode site-depth had a monotonic increase in the proportion of active sites as these channels go deeper into the cortical surface.

### 3.2 GRADIENT BOOSTING CAN PREDICT WITH HIGH PERFORMANCE THE PRESENCE OF ACTIVE CHANNELS

Table 1: Predictive modeling performance across classifiers. Numbers in parentheses represent 95% CI from 1,000 bootstrapped iterations.

| MODEL | SUBSET | ACCURACY | AUC-ROC | F1 SCORE | PRECISION | RECALL |
|---|---|---|---|---|---|---|
| Gradient Boosting | test | **0.73 (0.68-0.79)** | **0.74 (0.67-0.81)** | **0.64 (0.54-0.72)** | **0.73 (0.63-0.84)** | **0.63 (0.52-0.72)** |
| Gradient Boosting | ext. val. | **0.73 (0.68-0.79)** | **0.79 (0.73-0.86)** | **0.79 (0.74-0.84)** | **0.83 (0.77-0.89)** | **0.76 (0.69-0.83)** |
| Log. Regression | test | 0.69 (0.63-0.75) | 0.72 (0.64-0.80) | 0.62 (0.52-0.70) | 0.66 (0.56-0.76) | 0.65 (0.55-0.75) |
| Log. Regression | ext. val. | 0.36 (0.3-0.42) | 0.49 (0.44-0.54) | 0.07 (0.03-0.12) | 1.00 (1.00-1.00) | 0.04 (0.01-0.06) |
| KNN | test | 0.62 (0.56-0.68) | 0.59 (0.50-0.68) | 0.54 (0.46-0.62) | 0.55 (0.46-0.64) | 0.60 (0.49-0.71) |
| KNN | ext. val. | 0.37 (0.30-0.43) | 0.58 (0.51-0.64) | 0.12 (0.06-0.18) | 0.88 (0.67-1.00) | 0.07 (0.03-0.10) |
| Decision Tree | test | 0.69 (0.64-0.76) | 0.65 (0.58-0.73) | 0.60 (0.50-0.68) | 0.65 (0.53-0.77) | 0.58 (0.48-0.7) |
| Decision Tree | ext. val. | 0.66 (0.60-0.72) | 0.66 (0.61-0.72) | 0.73 (0.67-0.78) | 0.79 (0.73-0.86) | 0.67 (0.60-0.75) |

Table 1 shows the results of our predictive modeling across different metrics for different classifiers. The average accuracy on the holdout test of the development set was 0.69 (95% CI: 0.63-0.75) for

logistic regression, 0.62 (0.56-0.68) for KNN, and 0.69 (0.64-0.76) for the decision tree classifier. On the other hand, the accuracy for the external validation was 0.36 (0.3-0.42) for logistic regression, 0.37 (0.3-0.43) for KNN, and 0.66 (0.6-0.72) for decision trees. In contrast, our gradient boosting model had the best accuracy (0.73) for both the test and external validation set. Gradient boosting also had a better overall performance among the test and external validation sets for other metrics. For instance, the test set had AUC-ROC and a F1 score of 0.74 (0.67-0.81) and 0.64 (0.54-0.72), respectively, while the external validation set had an AUC-ROC of 0.79 (0.73-0.86) and a F1 score of 0.79 (0.74-0.84). Altogether, the non-parametric nature of gradient boosting allows us to predict with high performance when a channel will have sortable units. One of the advantages of gradient boosting is that it is not as affected as other complex models in the typical trade off between performance and explainability (Lundberg et al., 2018). Hence, using SHAP's tree explainer, we decided to further investigate how gradient boosting reached these predictions in order to obtain global explanations and determine feature importance of this model.

### 3.3 GLOBAL EXPLANATIONS INDICATE FEATURE IMPORTANCE

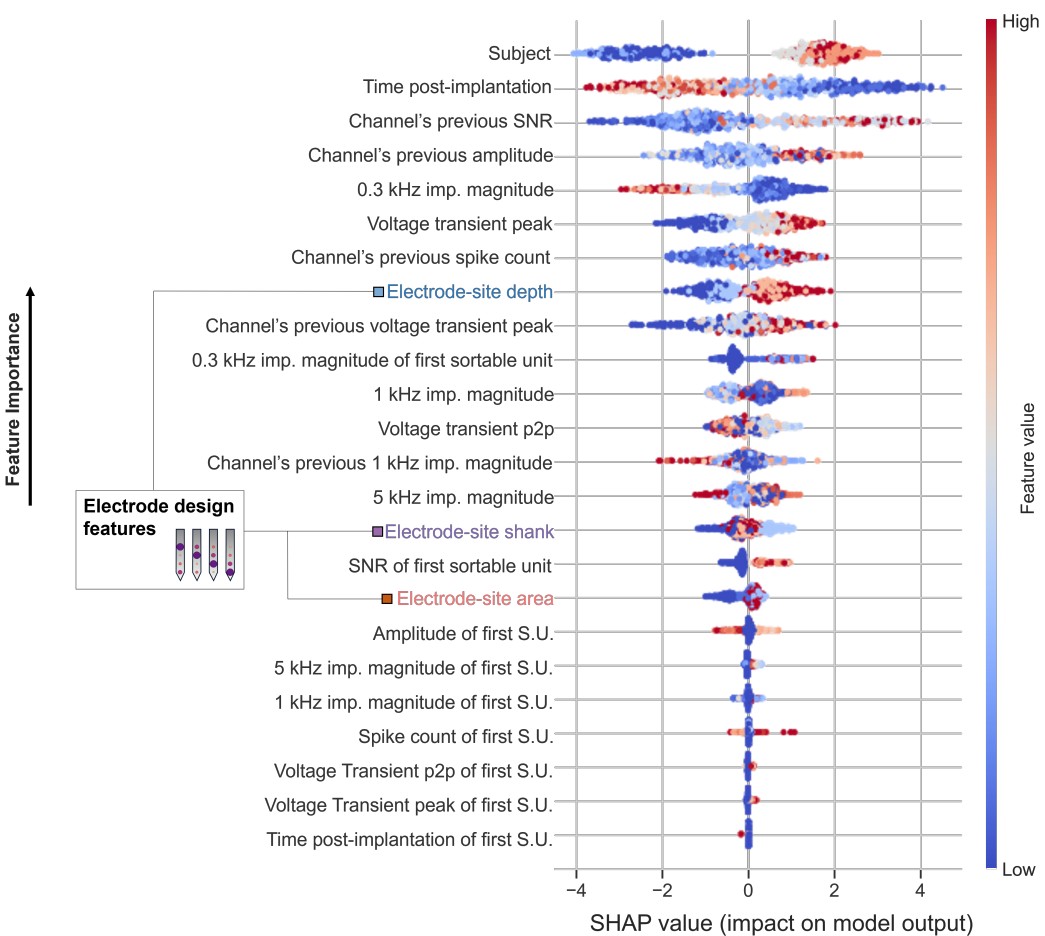

Figure 2: Summary plot containing global explanations for the most important features across the study. Feature importance is represented in descending order, with the most important features at the top of the graph. SHAP values indicate the overall contribution of an individual feature, higher values push the prediction towards the channel being active.

SHAP is a robust model-agnostic method to quantify the contribution of individual features for each prediction the model does (Lundberg & Lee, 2017). SHAP values are estimated by running a model with and without the feature in question while at the same time iterating the order in which the

feature is presented to reduce bias. This way, SHAP allows us to elucidate the feature importance from otherwise black box models and obtain global explanations and elucidate the most important predictors (Lundberg et al., 2018). Fig. 2 shows a beeswarm plot with the most important features for the prediction of active channels in our longitudinal recordings for BCI dataset. In this graph, positive SHAP values have a positive impact on the model output, in other words, they drive the prediction towards a channel being active (having sortable units), while negative SHAP values push the prediction towards the channel not being active. Our results showed that Subject ID had the greatest impact in our model. Interestingly, the distribution of SHAP values for this feature shows a bimodal distribution with some subjects having high SHAP values (red dots) while others having low SHAP values. This is consistent with the phenomenon of responders vs. non-responders observed across several BCI studies (Williams et al., 1999; Downey et al., 2018). Time post-implantation was the feature with the second largest impact in our model. Indeed, for the first half of the study 55.5% of channels were active while only 46.8% were active during the second half. The high feature values of Fig. 2 (dark red dots) had mostly negative SHAP values, suggesting that late timepoints tend to drive the prediction negatively (towards a channel not being active). This finding is consistent with the decay in the count of sortable units observed in chronic timepoints across the literature (Downey et al., 2018). Features based on the channel's previous sortable unit metrics (SNR, amplitude, and spike count) were among the top 7 most relevant features, with previous SNR and amplitude being the 2nd and 3rd most important features, respectively. The channel's previous electrochemical features such as VT peak and and 1kHz impedance magnitude, were ranked 9th and 13th for channels previous VT peak, respectively. These findings suggest that predictions rely heavily on the results of the previous experimental session, with channels with high SNR and amplitude in the previous recording having a higher probability of being active. Electrochemical characteristics such as 0.3 kHz impedance and VT peak were the 4th and 9th most important features. For the most part, low impedance values and high VT peak values seem to push the prediction towards the channel being active. Indeed, the average 1kHz impedance magnitude for inactive channels was 554.6 $\pm$ 468.3 s.d.and 480.7 $\pm$ 399.6 s.d. for active channels. Being ranked 8th, electrode-site depth was the electrode design variable that had the highest feature importance. Predominantly, red dots (light red: 700 μm, dark red 900 μm) had higher SHAP values, suggesting that deeper electrode-sites increased the probability of a channel being active. In contrast, both site shank and site area had similar contributions to the model but they are less important, with shank and area being the 15th and 17th most important feature in our model, respectively. Lastly, the electrochemical and spike metrics of the first sortable unit of a particular channel had an almost negligible importance (Fig 2, bottom). Many of the most important features of fig. 2 were consistently ranked among the top predictors across individual subjects and across classifiers. Collectively, the insights obtained from these findings, led us to further investigate how the model makes individual predictions on a specific recording session.

### 3.4 LOCAL EXPLANATIONS CAN SHED LIGHT INTO INDIVIDUAL CONTRIBUTIONS FOR A PARTICULAR RECORDING SESSION

SHAP's local explanations allow us to elucidate how decisions are made for individual instances, in other words, how each individual feature contributed to reach a prediction (Lundberg & Lee, 2017; Lundberg et al., 2018). To contrast how local explanations change for an active and an inactive channel, we selected a neural recording session on a chronic timepoint (week 10) for one of the animals (subject 2) of the development test. Subsequently, we picked 2 out of 16 channels at random, one with sortable units and one without them. Figure 3a shows local explanations for channel 5, an active channel. The axis units are log of odds, with higher values increasing the probability of a channel being active. Features in red increase the probability of a channel being active, while features in blue decrease it. Based on its magnitude (widest red bar), the most important feature driving the prediction positively was the previous SNR of the channel, which was 7.46. The other most important features in order of importance were the subject ID, the channel's previous amplitude, VT peak, and electrode-site depth. Conversely, time post-implantation was the most important factor driving the prediction towards a negative outcome. This is consistent with the observation of Fig. 2 in which late timepoints drive the prediction towards the channel not being active. Altogether, the additive nature of these explanations leads to a log of odds of 4.98, which corresponds to a probability of 0.99, in other words the model correctly classified this channel as active. On the other hand, Fig. 3b depicted local explanations for channel 10, a channel with no sortable unit. The most important factors pushing this instance towards a negative prediction were 0.3 kHz impedance magnitude, time post-implantation, and site depth. Interestingly, compared to the prediction of channel

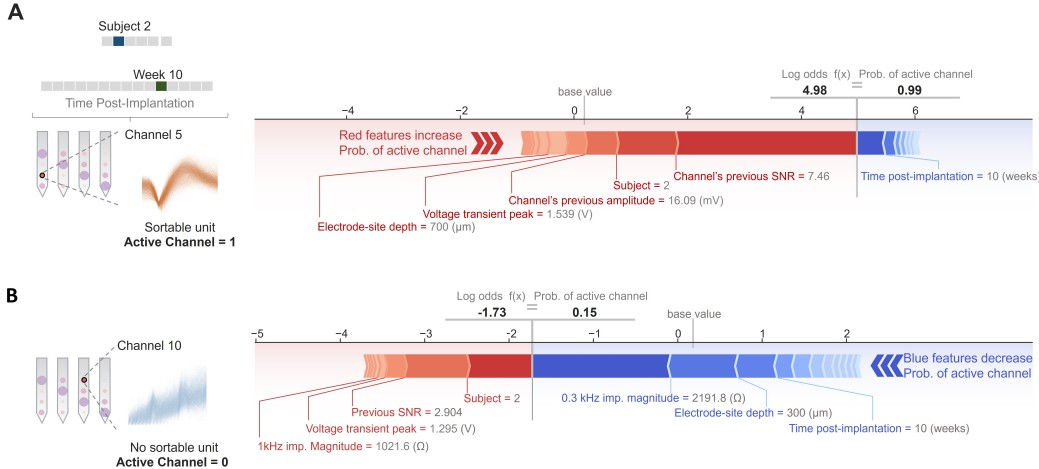

Figure 3: Local explanations depicting individual predictions for an active and an inactive channel for a single session of neural recordings. A) Local explanations for a channel with an active channel. Red features pushing the probability of a channel being active are more predominant on this plot. B) Local explanations for a channel with an inactive channel. blue features, decreasing the probability of a channel being active, are more prominent. The final prediction in the form of log of odds and probability are present at the top of each graph.

5 (channel with sortable unit) where site depth influenced the prediction positively, in this case, site depth had the opposite effect. This can be explained by the fact that the depth of channel 5 was 700 μm, compared to 300 μm for channel 10, the most superficial channel. As reported in Fig. 2, superficial channels predominantly had negative SHAP values, meaning that superficial channels (such as channel 10) tend to drive the prediction towards not having a sortable unit. The final prediction for channel 10 (Fig. 3) had a log of odds of -1.73, which corresponds to a probability of 0.15, in other words the algorithm correctly classified this channel as not active (0). The results of Fig. 3 showed that feature importance varied substantially from positive to negative predictions in just two instances. This raised the question: how does feature importance change across positive and negative predictions for more than two individual instances?

## 3.5 FEATURE IMPORTANCE ON CHRONIC TIMEPOINTS CHANGES DEPENDING ON WHETHER CHANNEL WAS ACTIVE OR NOT

To assess this question, we used decision plots for every negative and positive prediction for the second half of the longitudinal study. Similar to force plots (Fig. 3), decision plots can represent prediction paths for multiple individual predictions, allowing us to individually rank the most important features for active and inactive channels (Fig. 5). The x-axis is in log-odds, with higher values representing a higher probability that the channel is active. Each vertical line represents a single recording session for a channel. By following the path of individual lines across different features, we can determine how the model reached an individual prediction. Similar to Fig. 2, feature importance is sorted in descending order. Figure 5 (Appendix) a shows a decision plot for all inactive channels after the second half of the study. The most important factors leading to a negative prediction were subject ID, time post-implantation, channel's previous spike metrics (SNR, amplitude, spike count), and electrode-site depth. On the other hand, Fig. 5b shows individual prediction paths for active channels. In this case, the most important feature was the channels' previous SNR, followed by subject ID, and 0.3 kHz impedance magnitude. We saw a shift in the importance of time post-implantation. This factor went from the second most important feature for inactive channels (Fig. 5a) to the fifth most important feature for active channels (Fig. 5b). This shift in importance suggests that time post-implantation has a higher impact in the prediction of inactive channels, consistent with the observations of figures 2 and 3. Regarding features related with the design of the electrode, only electrode-site depth had relatively high importance for negative and positive predictions, 6th, and 8th, respectively. Other electrode design variables (site-area, and shank) were out of

top 10 most important features. Overall, analyzing decision paths for multiple predictions, help us elucidate intrinsic differences in feature importance leading to the prediction of sortable units for BCIs.

# 4    DISCUSSION

In this study, we recorded neural signals from rats chronically implanted with a custom-designed intracortical microelectrode. In order to better understand the long-term stability of these signals for BCIs, we developed an explainable ML pipeline to predict with high performance when a particular channel was active and to elucidate how different factors influenced these predictions. Using gradient boosting, our model had an AUC-ROC of 0.79 (95% C.I: 0.73-0.86) predicting whether a channel was active on unseen data. Through the use of SHAP, our pipeline identified that subject ID and time post-implantation were among the two important features in the model. This was followed by the spike metrics of the previous recording session. Importantly, out of the electrode design variables of our unique device, electrode-site depth was the most important, ranking 8th most important feature overall. Moreover, through the use of local explanations and decision plots, we were able to elucidate how the model made predictions on chronic timepoints for individual instances and how different feature importance varied based on whether the channel was active or not. For instance, time post-implantation played a more significant role in predicting whether a channel was not active than active. The significance and novelty of this study is two-fold: 1) it is a unique longitudinal dataset with a custom-designed microelectrode array. 2) To our knowledge, this is the first time predictive modeling and/or ML explainability tools have been applied to a chronic BCI dataset.

Our study had a few limitations and constraints. First, we limited our longitudinal study to 15 weeks only. Although this is a common timeframe for rats (Saldanha et al., 2021), human BCI studies might last several years (Hughes et al., 2021; Downey et al., 2018). However, animal and human recordings have shown similar trends over time (Downey et al., 2018; Williams et al., 1999). Therefore, this study might serve as a precedent for the implementation of ML tools on future human work. Another potential limitation is that for this study, we limited our dependent variable to the presence or absence of sortable spikes (single units). Some modern BCIs might rely on different neural signals using different filtering techniques. However, isolating single units remains an important metric for many BCIs (Lebedev & Nicolelis, 2017) and other neuroscientific research (Buzsáki et al., 2012). The feature importance of electrode-site area in our study might be limited to the design constraints of the microelectrode device. Even though our device had electrode-sites with different areas ranging from 800 μm² to 21,700 μm², the width of the shank was held constant. Many studies have shown that reducing the thickness and width of the device to subcellular levels ($\leq 10$ μm) can improve both the biocompatibility and chronic stability of its recordings (Luan et al., 2017). In conclusion, through the design of a custom microelectrode device and the development of an explainable ML pipeline we were able to predict with high performance whether a channel was active for BCI applications. This pipeline also allowed us to shed light on the most important factors in the model including the roles of inter-animal variability, time post-implantation, electrochemical features, and electrode design variables. Even though ML explainability tools such as the ones used in this study do not imply causation, our findings can inform the design of electrode manufacturing and future BCI studies. Similarly, this study can open the doors for the implementation of similar ML and explainability techniques to the field of neural engineering. Combining high performance predictions with explainability tools can help both clinicians and researchers alike as well as improving the long-term performance and wide implementation of BCI assistive technologies for millions of patients with paralysis.

# 5    ETHICS STATEMENT

This study involved handling, surgical implantation and experimentation of Sprague-Dawley rats. The study was reviewed and approved by the Institutional Animal Care and Use Committee (IACUC) of the last author's institution.

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

## 6  APPENDIX

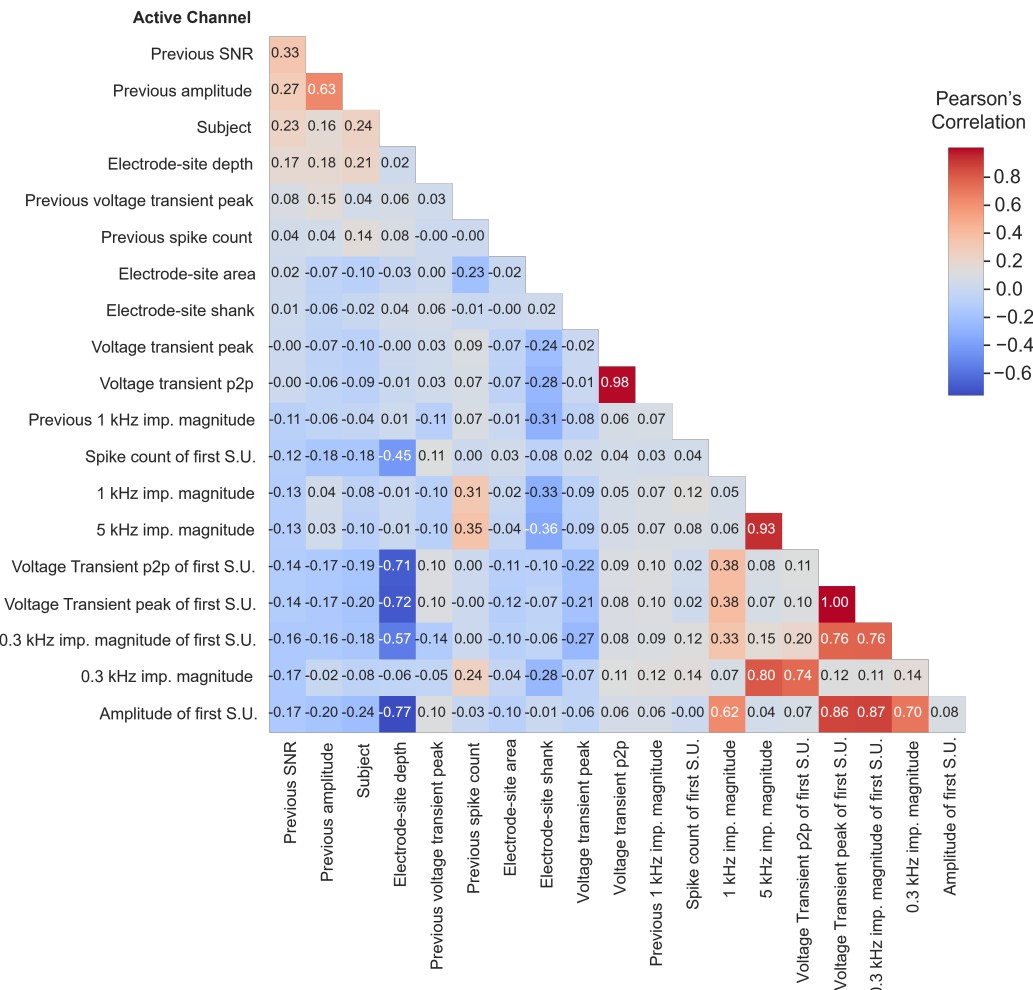

Figure 4: Cross-correlation matrix across features. Pearson's correlation value for each model variable and the binary dependent variable: active channel. The diagonal of this matrix (1) has been hidden for simplicity. S.U.: Sortable units.

Table 2: Model's input features: S.U. stands for sortable units

| FEATURE | DESCRIPTION | UNITS |
|---|---|---|
| Subject | Subject ID | Int |
| Time post-implantation | Time after the animal was implanted | weeks |
| Electrode-site depth | Cortical depth of the electrode site | µm |
| Electrode-site shank | Shank position in the array (1 most anterior - 4 most posterior) | Int |
| Electrode-site area | Size of the electrode-site | µm² |
| Channel's previous SNR | Signal-to-noise ratio of the last S.U. a particular channel had | Float |
| Channel's previous spike count | Count of spikes of the last S.U. a particular channel had | Int |
| Channel's previous amplitude | Peak-to-peak amplitude of the last S.U. a particular channel had | µV |
| Channel's previous 1 kHz imp. mag. | 1 kHz impedance magnitude of the previous experimental session | Ω |
| Channel's previous VT peak | Voltage transient peak of the previous experimental session | µV |
| 0.3 kHz imp. magnitude | Impedance magnitude frequency response at 300Hz | Ω |
| 1 kHz imp. magnitude | Impedance magnitude frequency response at 1kHz | Ω |
| 5 kHz imp. magnitude | Impedance magnitude frequency response at 5kHz | Ω |
| Voltage transient peak | Voltage transient peak of the current experimental session | µV |
| Voltage transient p2p | Voltage transient peak-to-peak of the current experimental session | µV |
| Time post-implantation of first S.U. | Week in which a particular channel had its first S.U. | weeks |
| SNR of first S.U. | Signal-to-noise ratio of the first S.U. registered for a channel | Float |
| Amplitude of first S.U. | Peak-to-peak amplitude of the first S.U. registered for a channel | µV |
| Spike count of first S.U. | count of spikes of the first S.U. registered for a channel | Int |
| 0.3 kHz imp. magnitude of first S.U. | Impedance at 300Hz of the first S.U. registered for a channel | Ω |
| 1 kHz imp. magnitude of first S.U. | Impedance at 1kHz of the first S.U. registered for a channel | Ω |
| 5 kHz imp. magnitude of first S.U. | Impedance at 5kHz of the first S.U. registered for a channel | Ω |
| Voltage Transient peak of first S.U. | Voltage transient peak of the first S.U. registered for a channel | µV |
| Voltage Transient p2p of first S.U. | Voltage transient peak-to-peak of the first S.U. registered for a channel | µV |

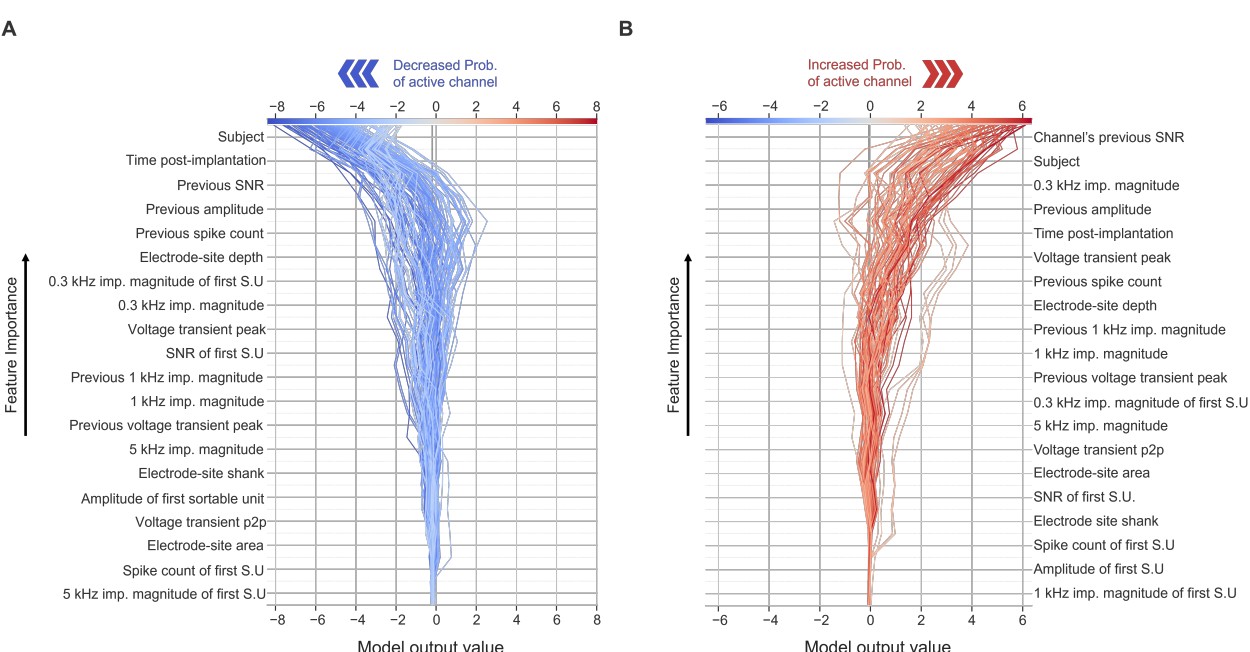

Figure 5: Decision plots on chronic time points for predictions of channels with and without sortable units. All individual trials of the second half of the study were considered for this analysis. Feature importance increases from bottom to top. A) decision plot showing prediction paths for every inactive site. The lower the model output value the lower the probability that a channel has a sortable unit. B) decision plot showing prediction paths for every active site. The higher the model output value the higher the probability that a channel has a sortable unit. S.U.: sortable units.

