# OpenReview forum: "Explainable Machine Learning Predictions for the Long-term Performance of Brain-Computer Interfaces"
_ICLR.cc/2023/Conference — Submitted to ICLR 2023_

### Official Review · Reviewer_byYz · 2022-10-22

**Confidence:** 4
**Correctness:** 4
**Technical Novelty And Significance:** 4
**Empirical Novelty And Significance:** 4
**Recommendation:** 8

**Clarity, Quality, Novelty And Reproducibility:**

This is certainly one of the strengths of the paper. Except for a couple of aspects that I would like to ask for more detail, but certainly this paper scores high on all; clarity, quality, novelty, and reproducibility. Regarding that couple of aspects:
+ Why is the 6th rat considered useful for “external validity” yet there Is no evidence or explanation that this rat would be statistically different from the first 5, and certainly the surgery date is not a factor affecting the neural responses of the rodent, is it? If the rat is then statistically similar to the others (besides the natural inter-subject variability for which the other rats may be as different for all that I understand) then there is no reason to accept the 6th subject as useful for claiming external validity and it might be better to use it to increase the internal validity.
+ 1000 replication for 95% statistical power seems pretty large even for small effect sizes. No further details of the power analysis is given. Can the authors indicate how was this calculated? Is perhaps the study overpowered?


**Details Of Ethics Concerns:**

Animal models are used for experimentation. Moreover, an invasive procedure is conducted (to implant the electrodes). However, no ethical approval details are indicated.

**Strength And Weaknesses:**

Strengths

+ The problem studied is clear and remains open and the study proposed in this draft is a clear contribution to the field.
+ The introduction is entertaining and of adequate depth, the literature review is extensive and its use throughout the article is excellent, the methodology is solid, the results are clear, the discussion (embedded within the results) is interesting, and the article has high nomological validity.

Weaknesses

Not many…
+ Absence of inferential statistics, and even the descriptive statistics is incomplete. For example, the standard deviations of the different statistics are not described, the statistical power analysis is not described, etc.


**Summary Of The Paper:**

The problem tackled is that of the decay in performance of BCI systems over time. The article presents an analysis of an electrocorticography data set (n=6) used for brain-computer interface, and in particular discusses the importance of the different features used by a predictive model built with machine learning techniques.
Four different models are put to the test and the one exhibiting highest prediction rates is further analyses to decode the importance of the different features.
Although ML is popular for BCI applications, I agree with the statement that for the specific case of boosting longitudinal stability of the BCI its application is novel.


**Summary Of The Review:**

+ Without an analysis of error propagation it is difficult to anticipate how far in the long term would the predictive power of the model hold within useful boundaries.
+ There appears to be no measure of the degradation of the performance of the xML pipelines as the prediction horizon is further away.
+ Cross-correlated variables were removed. Were corrections considered (as opposed to simply dropping) e.g. statistical whitening, Bayesian multivariate decoupling, ARIMA models with a cross-variate feed, etc?
+ The explanatory approach methods taken here (SHAP and decision paths) is actually very insightful and in principle is generic, hence it could be translated to other BCI modalities; e.g. EEG based, fNIRS based, etc. What adaptations would be suggested to do so?
+ Although the translation to humans is identified as one of the limitations of the study, I can see how rather than a limitation of this study that makes up for a different study altogether. In this sense, my question is a bit more humble, regardless of the length of the BCI intervention, and focusing strictly on the predictive horizon of the model (currently shown at 15 weeks for the rats), is it likely to be at least equally long in humans? Even if it is only for 15 weeks, that remains a quite reasonable horizon in humans, but unfortunately, with a much more complex nervous system, I reckon the decay of the performance of the predictive models might suffer more heavily?.


Minor details
+ Figures looks grainy on my end (although readable). Not sure whether this is only on my side, but just in case it does not cost much to double check.
+ AI and ML are closely related but not the same. The authors seem to use explainable machine learning and explainable artificial intelligence interchangeably. If this is intentional, I’m fine with it. If not, why not stick to only one of them? Subjectively I would say this paper in particular is closer to xML rather than xAI but I wouldn’t argue with other perceptions.
+ Some of the things explaining the figures are in the main text. Of course, every author has its style and I do not want to interfere. But if there is no strong preference, may I suggest to move such parts to the figure caption to make the images self-contained?

---

> ### Author Response · Authors · 2022-11-16
> **We thank the reviewer for such a thorough review and great feedback.**
>
>
> Statistics:
>
> Thank you for this suggestion. We have expanded the statistics of different features in the results section. For instance, we described how the proportion of active channels shifts between the first and second halves of the study as well as the average impedance magnitude for active vs. inactive channels. We have also included a supplementary table to describe each of the features used in the model in more depth.
>
> Using the 6th subject as an external validation set:
>
> We concur with the reviewer. Even though this subject was from a different cohort (a few months younger) and was not habituated to the recording chamber for the same amount of time as the previous 5, we do not expect this animal to be significantly different from the rest. As the reviewer suggested, we could have added this subject to the training cohort to increase the internal validity set. However, in order to emulate a real-life scenario and improve the translatability of this study, we used this subject as the external validation set so we can compare the performance of our algorithm on unseen data.
>
> 95% confidence intervals calculation:
>
> We calculated the 95% confidence intervals from the resampling distribution by using 1000 bootstrap resamples. This type of 95% C.I. calculation is known as the empirical C.I. calculation (see: DiCiccio and Efro, 1996). We implemented this through the use of the percentile() function of the NumPy library. Lastly, the choice of 1000 bootstraps is commonly used in the field to obtain the confidence intervals of performance metrics (i.e. Thorsen-Meyer et al. 2020).
>
> Removed variables:
>
> We removed predictors that were not available prior to obtaining the neural recordings to improve the clinical relevance of the study. Similarly, we also excluded potential variables that might not be available in clinical BCI studies. After this process, the only remaining correlated variables were those with related units such as day/week/month post-implantation.
>
> Translating our machine learning pipeline beyond intracortical BCIs:
>
> We thank the reviewer for pointing this potential application of our methods. As suggested, it would not require too many adaptations; it would only require a change in the type of features added to the training set. For instance, by shifting from intracortical BCI to EEG, we would replace the variable "Electrode-site depth" by the EEG channel location/coordinates. We have made changes to the discussion section to describe the potential application of these techniques for longitudinal datasets beyond intracortical BCIs.
>
> Translation to humans:
>
> This is an interesting point. There are some human studies that have lasted for several years (i.e. Hughes et al. 2021). The electrodes used in human intracortical BCIs typically have 96 channels, compared to the 16 channels used in this study. In theory, this would mean more training data, which would make the model more robust. Similarly, there are interesting human-specific features that can be potentially added to the algorithm, such as the emotional state of the patient, which can be measured via heart rate, blood pressure, and skin conductivity. Altogether, with a dataset large enough, we hypothesize that our algorithm would be more robust for human BCIs.
>
> Figure quality:
>
> We thank you for the advice. We exported the figure at the maximum quality available, but we'll make sure the figures are not pixelated in the camera-ready version.
>
> Explainable machine learning and explainable artificial intelligence:
>
> We thank the reviewer for bringing this up. We will follow the reviewer's recommendation and stick with "Explainable machine learning" throughout the manuscript.
>
> Figure captions:
>
> Thank you for this recommendation. We have expanded the caption of figures. We are positive that this change will increase the legibility and flow of our manuscript by having self-contained figures.
>
> Ethics Statement:
>
> Thank you for pointing this out. We have updated the manuscript with a section for the Ethics Statement. This statement indicates the use of animals in the study and includes the approval of the pertinent entity.

---

### Official Review · Reviewer_eCAt · 2022-10-25

**Confidence:** 5
**Clarity, Quality, Novelty And Reproducibility:** this work is clear and novel.
**Correctness:** 3
**Technical Novelty And Significance:** 3
**Empirical Novelty And Significance:** 4
**Recommendation:** 5

**Strength And Weaknesses:**

The collected data is very interesting and useful for the proposed task (predicting dead channels and explaining what factors influence dead channels). The analysis is sound and well-conducted.

However, I think that the claims made in the paper are a little too strong.

The claim, in the abstract, that "Predicting whether a channel will have sortable units across time would mitigate this issue and increase the utility of these devices by reducing uncertainty, yet to-date, no such methods exist" is not true. This has been done (see for example Spike sorting pipeline for the International Brain Laboratory, IBL, 2022), with a method predicting bad channels in Neuropixels probes based on similarity with neighboring channels. I agree that this is different from the work in this paper but it would be worth explaining the difference a little better.

Also, I think it would be straightforward to extend the author's analysis to high-density probe recordings where dead channels can be detected, to evaluate the accuracy of the prediction.

I would have liked to see a schematic of the experiments, showing a diagram of the probe (it is hard to know what the probe looks like from the text), and also explain the features that are input to the model in more depth. I couldn't understand if the authors use all the spikes info, and how they calculate SNR, amplitude or magnitude for the channels.

The novelty of the approach is that the model predicts dead channels using features like Subject/time post implantation, that have a high SHAP value. However, lots of additional features could have been used and the selection of features seems bit arbitrary. Why use channel's amplitude and SNR? Having a model that solely predicts the dead channels using info that is not contained in the signal would be much more impressive and informative. Moreover, I think it should be easy to get higher accuracy when using signal info.

Finally, it seems to me that "time post-implantation" is obviously important to predict dead channels, as well as "Subject" and it does not really inform us about the reason why a channel dies (i.e. Subject could mean many different things - such as a problem when implanting the probe or else), which make the conclusions a little less informative.





**Summary Of The Paper:**

The authors propose a new unique longitudinal dataset with a custom-designed microelectrode array implanted in 5 rats to develop models for predicting microelectrodes arrays dead channels and, more importantly, explain what factors are the most influential for predicting dead channels.


**Summary Of The Review:**

The idea presented in this paper is interesting and novel, but the model should be refined (i.e. by using different features for prediction and explaining the choice of features a bit more). The conclusions drawn from the model could potentially be much more informative using a different set of features. I hence don't recommend acceptance of this paper.

---

> ### Author Response · Authors · 2022-11-16
> **Part 1 - We thank the reviewer for the comments. We have incorporated changes to the manuscript that will avoid future confusion**
>
>
> Number of animals implanted:
>
> We want to clarify to the reviewer that we implanted a total of 6 animals. However, one of these was used as the external validation set during leave-one-out cross validation (see Figure 1 Implantation & Model development sections).
>
> Predicting whether a channel will be active over time:
>
> The reviewer suggests that the task of predicting if a channel will be active over time has been done before and provides the 2022 publication by the International Brain Laboratory (IBL) as an example. We thank the reviewer for this comment. However, after carefully examining the 2022 publication, there are a few fundamental differences between the methodology of that paper and our study: 1) The IBL paper describes a spike sorting method, not a predictive modeling pipeline on a longitudinal dataset. According to the cited manuscript, when a channel is inactive (dead) it replaces the voltage of the inactive channel with the voltages of contiguous channels using spatial interpolation (IBL, Banga et al., 2022). Even though this interpolation technique has great implications for the field, it is completely different from our methodology. 2) The voltage interpolation is done after obtaining the neural recordings. This is distinct from the aim of our study. ML pipeline predicts whether a channel will be active, prior to obtaining neural recording for a given session. This is why we only use spike data from the "channel's previous" recording sessions. From the translatability point of view this is very significant since these predictions can be included into a decoder prior to the experimental session, potentially reducing or even eliminating recalibration time (recalibration is one of the key factors hindering the translation of clinical BCIs; see: Blabe et al., 2015). 3) The IBL paper does not mention any prediction (or even interpolation) of active/dead channels over time. 4) In contrast to the spike sorting pipeline of the IBL paper, we do not use raw neural data as input in our algorithm (Figure 4, table 2). 5) The reason why it is possible to interpolate (not predict) voltages based on neighboring channels in this IBL paper is because of the dimensions of the NeuroPixel probe. The NeuroPixel device has an inter-electrode site distance ~20 µm (Pitch of 16 µm x 20 µm and site size of 12 µm x 12 µm). In contrast, the inter-electrode site distance of our custom-made probe is an order of magnitude larger (see Figure 1). This inter-electrode distance is more similar to the Utah microelectrode array (inter-electrode distance of 400 µm) which is the only FDA approved intracortical device in human studies. 6) Compared to our custom-made electrodes, the NeuroPixel probes do not have variations in many dimensions (e.g., site size, site depth, shank location), which makes it unsuitable for testing the importance or effect of these design variables. In order to avoid potential confusion by future readers, we have changed the wording of the abstract from "to-date" to "to our knowledge". We also included the dimensions of the custom-made device and explicitly mentioned that the predictions are pre-recordings in Figure 1 and in the methods section.
>
> Extend the author's analysis to high-density probe recordings:
>
> We think that applying similar ML pipelines to longitudinal BCI datasets has great potential to propel the field forward. Indeed, we suggest this in our discussion: "This study can open the doors for the implementation of similar ML and explainability techniques to the field of neural engineering". We envision that "this study might serve as a precedent for the implementation of ML tools on future work" (see: discussion section). We agree that even if the dimensions of the NeuroPixel probe are orders of magnitude smaller than clinically approved BCI electrodes (i.e. Utah array), this experiment would still have a great impact for the neuroscientific community. However, implanting new subjects, collecting longitudinal neural recordings, and applying our ML pipeline on a different electrode is beyond the scope of this manuscript.
>
> Schematic of the experiments and diagram of the device:
>
> Figure 1 contains a descriptive diagram of our experimental pipeline. We have now included the dimensions of the device into the figure (implantation) to improve the clarity of the manuscript. We have also included more details into the pre-processing and model development sections of the figure and extended the figure caption. We thank the reviewer for this suggestion.

---

> > ### Author Response · Authors · 2022-11-16
> > **Part 2 - We thank the reviewer for the comments. We have incorporated changes to the manuscript that will avoid future confusion**
> >
> >
> > Explain the features that are input to the model in more depth:
> >
> >  We have included a table (Table 2) that includes a description and the corresponding precision and units of each feature used by the ML algorithm. This is an extension of supplementary Figure 1 (see Appendix). Information that we include from a channel's previous sortable unit (number of spikes, amplitude, SNR) come from the Plexon Offline Sorter. We have edited the “Variables and feature engineering” subsection to incorporate this information.
> >
> > Features selection:
> >
> >  We focused on features that were available prior to recordings and that are commonly used in clinical BCI applications. For example, we used spike and electrochemical features of the channel’s previous sortable unit as well as metrics of the first time the channel was active. We also use electrochemical data such as voltage transient and impedance magnitude (at commonly used frequencies). These are techniques used in the field to understand the electrochemical properties at the electrode-tissue interface and are routinely collected prior to human BCI experiments (Colachis et al. 2021, Flesher et al. 2016). The only features in this experiment that are different from those seen in BCI research and clinical applications is the combination of site area, site size, and site depth on our single custom-made microelectrode array. Common microelectrodes tend to have a single value of these design variables. Our device has a range of site areas, site sizes, and site depths. The range of these variables on our device encompass those seen in research and clinical devices. Thus, our device can be thought of as a testbed for understanding the interplay of these design parameters, and our results can then inform the choice of these parameters for devices used in research and clinical settings.
> >
> > Given the wide variety of electrode designs that exist in the field and the heterogeneous responses that have been reported in the literature, we leveraged our custom-design with our explainable machine learning pipeline to tease out the most important factors in the long-term stability of intracortical recordings, including the electrode-design variables described above. We have made changes throughout the manuscript to better inform these distinctions. Additionally, table 2 (appendix) includes individual descriptions of the features used in the model.
> >
> > "Having a model that solely predicts the dead channels using info that is not contained in the signal would be much more impressive and informative":
> >
> > Our model does just this. It solely predicts active/dead channels using information that is not contained in the neural signal (it does include information neural signal information from the past). The input data for our predictive pipeline includes electrochemical features (pre-recordings) and intracortical recordings from previous experimental sessions to predict if a channel will be active or inactive. We also agree with the reviewer; there is an increase in accuracy when we provide spike data from the same experimental session (indeed, we did this experiment), but we believe this is not a particularly significant or novel finding as several other algorithms exist to determine if a channel is active given spike data at current time. However, the results presented herein come solely from spike data recorded in prior  experimental sessions because of its clinical significance. We have made edits to the overview subsection of the methods, parts of the introduction, and Figure 1 to explicitly mention this distinction.

---

> > > ### Author Response · Authors · 2022-11-16
> > > **Part 3 - We thank the reviewer for the comments. We have incorporated changes to the manuscript that will avoid future confusion**
> > >
> > > Adding time post-implantation and subject as predictors:
> > >
> > > We concur with the reviewer in that there is a large amount of literature reporting the effect of time post-implantation on the decay of neural signals and BCI performance. Indeed, we extensively talk about this phenomenon in our introduction and discussion sections. Similarly, inter-subject variability has been widely documented in the literature, where even in the same study some subjects are categorized as responders vs. not responders (i.e. Williams et al., 1999). For these reasons, time post-implantation and subject cannot be excluded from a predictive modeling perspective. In fact, these two features represented the most important predictors in our model. In the case of a longitudinal BCI study in humans, excluding the time post-implantation and patient ID would considerably decrease the performance of the algorithm, and hence the utility of the results. Moreover, from the point of view of explainable ML, even if the effect of these features might sound evident, they should not be excluded. In the case of global explanations of the explainable ML pipeline, these should include the same features used in the predictive modeling part. Additionally, a more in-depth analysis of these explanations can reveal interesting trends. For instance, in the case of chronic timepoints (Figure 5). We found that for active channel predictions (active channel = 1), the channel's previous SNR plays a more important role than time post-implantation. However, it is still the most important predictor for inactive channels (Figure 5). This unexpected shift would have not been possible if we censored features that could have been considered "obvious". Lastly, if an ML pipeline like ours were to be adopted in a research or clinical setting with the goal of predicting whether a channel will be active, the time post-implantation and the unique patient identifier would presumably be readily available information. Given this information 1) is largely supported by the literature as having an effect on the decay of neural signals (related to the outcome of our study) and 2) is readily available in the settings that a pipeline such as ours may be adopted, we strongly believe it is important to include in our study.

---

> > > > ### Comment · Reviewer_eCAt · 2022-12-07
> > > > **Re: Response to Reviewer eCAt**
> > > >
> > > > I thank the authors for the detailed response and the clarifications.
> > > >
> > > > I understand that the model only uses information that is not contained in the "current" neural signal but uses information from the past neural signal information. This is what I meant in my review, sorry if my comments were confusing. My comment about the the spike sorting paper was that we can basically predict all dead channels using neural signal - so basically if we use this to see what channels were dead in the previous signal, it could already give a good fraction of dead channels (as it would detect all channels that were dead before). Does this make more sense?
> > > > I am wondering if most of the predicted dead channels were dead already int he previous neural signal - This would be interesting info to add in the paper.
> > > >
> > > > The other thing is that if the model mostly relies on previous neural signal to predict dead channels, it won't give any insight into why the channel is dead - although this is one of the goal of this study. That's why I suggested to remove any past neural signal information in the study (or maybe simply quantify how much the performance decreases when not using past neural signal as a predictor)
> > > >
> > > > Overall, the authors have answered some of my questions and I believe the paper is a bit more clear now. The authors have clarified the importance of their work, and I believe the dataset is of great interest. However, I am still not convinced by the model as it is not clear what additional insight is revealed by the model.
> > > > I increased my score accordingly

---

### Official Review · Reviewer_Mjwm · 2022-10-26

**Confidence:** 3
**Correctness:** 3
**Technical Novelty And Significance:** 2
**Empirical Novelty And Significance:** 4
**Recommendation:** 6

**Clarity, Quality, Novelty And Reproducibility:**

Well written paper overall.

The text is not clear on how global explanations and local explanations are related, and how they differ.
Similarly, it is not clear what question is being answered in Section 3.5.

**Strength And Weaknesses:**

I am not sure how this paper fits into a typical ICLR paper, bu this paper addresses an important problem in BCI.

In addition to the SHAP values associated with each parameter, it might be helpful to see the impact of each parameter directly on the % of active channels (similar to what is done for electrode design parameters in Section 3.2). This would help understand how the methods used in this paper help with better explainability.

Stability of SHAP values across different partitions of individuals would be useful to know. That is, how uniform are the contributions of different features across different individuals?

Do the factor contributions (SHAP values) change with classifier type?

**Summary Of The Paper:**

This paper uses explainable machine learning methods to study which factors impact the number of active channels in a BCI. The authors analyse a number of important factors such a intersubject variability, time post-implantation, electrochemical properties and electrode design. The analysis relies on a reasonable dataset in mice.

**Summary Of The Review:**

While more analysis can be done, I think the paper is a nice contribution to an important BCI question. I am not sure how much this aligns with the theme of ICLR.

---

> ### Author Response · Authors · 2022-11-16
> **Thank you for the great suggestions. We have included statistics and more details on the model’s features.**
>
> More detailed analysis of features:
>
> We thank the reviewer for the suggestion. We have expanded on the statistics of the predictors in relation with the dependent variables. For example, we now describe how the percentage of active channels shifts between the first and second halves of the study. Moreover, we included a table in the appendix with detailed descriptions (including units) of each of the features used in the model.
>
> Stability of SHAP values:
>
> A feature importance comparison across partitions of subjects showed comparable results to those described in section 3.3 (Figure 2). For instance, features such as time post-implantation, electrode-site depth, and spike metrics of the last time the channel had sortable units (i.e., "channel's previous SNR") were among the top predictors across all subjects. A similar trend appeared when comparing XGBoost with other algorithms. For instance, time post-implantation, subject, channel's previous SNR, and electrode-site depth were the top predictors for the logistic regression model. We thank the reviewer for this recommendation. We have expanded our results sections to describe these findings.
>
> Insights from section 3.5:
>
>  Compared to the global explanations of section 3.3, the decision plots of section 3.5 allows us to elucidate the differences in feature importance for positive and negative predictions on chronic timepoints. For example, for negative predictions (channel will be inactive), time post-implantation is the important predictor (consistent with the results of section 3.3). However, an interesting shift occurs in the case of positive predictions, where the signal-to-noise ratio of the channel's previous sortable unit becomes the most important predictor and time post-implantation is relegated to the fifth position.
>
> Contributions of the manuscript and relevance to the conference:
>
>  We thank the reviewer for highlighting the importance and clinical significance of our manuscript to the fields of neural engineering and brain computer interfaces. Our study represents an important application of machine learning techniques to the field of neuroscience, which is one of the relevant topics of the conference. In fact, our longitudinal BCI dataset with a custom-made probe is completely novel and has never been published before. Additionally, our predictive modeling pipeline combines state-of-the-art machine learning algorithms with explainable ML techniques which are in concordance with the cutting edge research that is characteristic of ICLR papers.

---

### Official Review · Reviewer_9zRr · 2022-10-27

**Confidence:** 4
**Correctness:** 3
**Technical Novelty And Significance:** 2
**Empirical Novelty And Significance:** 3
**Recommendation:** 3

**Clarity, Quality, Novelty And Reproducibility:**

Clarity
- Overall paper is well written and well illustrated.

Quality


Novelty
- Novel dataset. Authors apply previous methods in the data analsyis analysis. Authors claim that "no ML studies have been applied on the longitudinal stability of intracortical BCIs." Custom-designed intracortical microelectrodes appears to be novel, but beyond the scope of the conference.

Reproducibility
- Reviewer is unable to comment on the details of the animal study/data generation. Details are provided to reproduce the data analysis.




**Details Of Ethics Concerns:**

Authors need to state the necessary approvals that were obtained for the animal study.

**Strength And Weaknesses:**

Strength
- Well designed study with novel dataset that is highly relevant to the neuroscience/BCI/machine learning community.

Weaknesses

- It is not clear that the paper answers the question of when a channel will become inactive or assesses the longitudinal behaviour of channels over time ("Predicting when stability of neural signals will decrease and understanding which features of the interface contribute to this, would benefit clinicians and researchers and improve utility of intracortical BCIs.")

- Authors do not provide results demonstrating how predicting whether a channel is active, and consequently the proportion of active channels is related to the downstream BCI task (e.g., movement of robotic arm). Assuming the proportion of active channels can be predicted, how is this information relevant to predicting the longitudinal performance of the BCI task?

- Not really clear what additional insight is revealed by some of the feature analysis (e.g., sections 3.4 and 3.5) and based on previous analysis in the literature (section 3.3, e.g., intuitively, the longer the implant duration, the higher probability of inactive channels due to build-up, high impedance, etc.).

- Small sample size and short duration of study, but understandable given the nature of the experiment. Authors acknowledge limitations.


**Summary Of The Paper:**

This paper focuses on invasive brain-computer interfaces (BCIs) using intracortical electrodes that are susceptible to degradation of signal quality over time. The goal of the study is to predict whether channels in intracortical electrodes are active based on features from neural recordings, electrochemical data and physical characteristics of the electrodes and assess feature importance. Results are presented from an animal experiment.

**Summary Of The Review:**


Recommendation is based on the strengths and weaknesses described above.

Other Comments:

-    While Figure 1 is well-annotated, caption should be more informative and direct reader through understanding the various components of the figure. Same for Figure 4.

-	Offline Sorter: Authors need to provide more details about Offline Sorter because it is confusing. Is it a software algorithm than can automatically identify spike waveforms or does it require manual review with “blind reviewers”? If Offline Sorter is automated, why is a ML algorithm need to predict active channels if similar information can be obtained using Offline Sorter?

-	Custom-designed intracortical microelectrode: Are some of the features able to be recorded automatically using conventional electrodes or are they available due to the custom designed microelectrode?
      o	 "First, we obtained the electrochemical (EIS and VT) and spike metrics of the first recording session with that had a sortable unit for a particular channel (”First S.U.”)."

---

> ### Author Response · Authors · 2022-11-16
> **Part 1 - We thank the reviewer for such a thorough review and for catching a typographical error in the methods section. We fixed this error and made changes to the manuscript to clarify many of the points brought up by the reviewer.**
>
> Assessing longitudinal behavior of channels:
>
> Our machine learning pipeline can classify with high accuracy whether a channel will be active (1) or inactive (0) at any moment across the duration of the study (15 weeks). Importantly, this can be done prior to obtaining the neural recordings. For example, by feeding the algorithm features such as time post-implantation, impedance (routinely gathered prior to experimentation in clinical settings), and the spike characteristics of the previous time a particular channel had sortable units, the algorithm can predict with high performance whether that particular channel will be active or not. In other words, we can predict the longitudinal stability of individual channels, which will influence their inclusion/exclusion in an optimal BCI, based on prior performance and impedance measurements that are necessary to ensure device fidelity. We have modified several aspects of the manuscript to make this more evident (see comments below for details).
>
>
> Relationship between recording stability and BCI performance:
>
> Neural signals coming from implantable electrodes represent the main source of information used (i.e., decoded) by intracortical BCIs to control external actuators (e.g., a robotic arm or a mobile phone).  Decoding performance is tightly associated with the presence and quality of these signals. Studies have shown that even very small changes in neural recordings---such as a single neuron changing identity over time---can result in decoder degradation (Downey et al, 2018). In our study, in our study, we address the common problem of a recording electrode losing signal from the neuron completely (i.e., channel becoming inactive). Channels with no viable neural signals introduce stochastic noise to the decoder and features derived from these channels may negatively impact decoder performance (Colachis et al. 2021). Therefore, an ML pipeline for the prediction of active channels can prevent this issue by identifying when a channel should be excluded from the BCI to maximize performance.
> When a channel becomes inactive, BCI performance can be salvaged by recalibration. However, the need for frequent recalibration is one of the key factors hindering the translation of clinical BCIs (see: Blabe et al., 2015). Hence, combining predictive modeling techniques similar to this study with unsupervised recalibrating decoders (i.e. Bishop, et al. 2014) can greatly benefit the field. Therefore, the key finding of our approach is not the proportion of active channels for a particular experimental session, but the ability to predict which channels will be inactive and more importantly the confidence with which these can be fed into the decoder. This study serves as a precedent for the use of predictive modeling and ML explainability tools for longitudinal BCI datasets. Hence, testing longitudinally how this information can be used to improve decoding performance is outside of the scope of this manuscript.
>
>  Single instance and decision plots on chronic timepoints:
>
> The global explanations of the longitudinal model representing the most important features of the model (Section 3.3 & Figure 2) contain the most relevant and summarized results of the manuscript. However, the results of sections 3.4 and 3.5 offer important insights that expand on the results observed in section 3.3. For instance, in the case of section 3.4, knowing how individual features contributed to reach a prediction for a particular channel can be very informative on how the model works. These insights may also be useful to BCI researchers hoping to optimize the device itself (i.e., which features of the neural implant maximize the likelihood of maintaining active channels). Additionally, there are interesting insights from this experiment like those shown in Figure 3, where we observe how a superficial channel can negatively impact the model (Blue: pushing the prediction towards the channel being inactive) (Fig. 3B) while a deeper channel can push the model in the opposite direction (Fig 3A: red). Similarly, we see a comparable phenomenon in the results of section 3.4, where time post-implantation on chronic timepoints shifts from being the second most important variable for the prediction of inactive channels to the 5th one for the prediction of active channels (Figure 5). Also, the SNR of the channel's previous sortable unit (and not time post-implantation or subject) is the most important predictor that the channel will be active on chronic timepoints.
>
> Figure captions:
>
>  We thank you for the recommendation. We are sure that these changes will improve the legibility and flow of the manuscript. In order to achieve this with the page limit constraints, we moved the figure of section 3.4 to the appendix.

---

> > ### Author Response · Authors · 2022-11-16
> > **Part 2 - We thank the reviewer for such a thorough review and for catching a typographical error in the methods section. We fixed this error and made changes to the manuscript to clarify many of the points brought up by the reviewer.**
> >
> > Clarification on the use of Plexon Offline Sorter software:
> >
> > With over 1,700 publications, Plexon Offline Sorter is one of the most popular methods to sort spikes in the field. However, it is important to note that Offline Sorter does not provide any prediction of active channels; rather, it is a data analysis software package. Neural data coming from freely behaving subjects is riddled with motion artifacts. Current automated sorting algorithms have a hard time sorting single units when the recordings have many motion artifacts. Therefore, the sorting process in Offline Sorter is done semi-automatically by allowing a user to manually remove signals that are clear noise or coming from motion artifacts. The blind reviewers were used to determine if the isolated spikes resembled an extracellular spike waveform and not the result of random noise or motion artifacts. We modified Figure 1 and the methods section to clarify this issue.
> >
> > "If Offline Sorter is automated, why is a ML algorithm need to predict active channels if similar information can be obtained using Offline Sorter?":
> >
> >  Offline sorter uses raw neural data as an input and sorts for spikes. Instead of sorting spikes from raw neural data, our ML algorithm predicts whether a channel will have sortable units (a sortable unit is a source of spikes in neural data) prior to obtaining these signals. This is a very important distinction since predicting whether a channel will be active immediately after feeding the model with impedance/VT data and other features available pre-recording has a much larger clinical relevance than the opposite scenario. We apologize for this confusion. We have edited Figure 1 to make this fact more evident and made changes throughout the manuscript to explicitly make this distinction.
> >
> > Custom-designed probe intracortical microelectrodes:
> >
> > Most of the features used in the model can be obtained from most intracortical microelectrodes. Features such as impedance are routinely obtained in clinical BCI applications. The only variables that are unique to this custom device are the combination of site area, site size, and site depth. Common microelectrode arrays do not include variations of these design variables on the same device. Our custom device allows us to test different combinations of these design variables in a device that is similar in all other respects to common microelectrode devices.
> > Given the wide variety of electrode designs that exist in the field and the heterogeneous responses that have been reported in the literature, we leveraged our custom-design with our explainable machine learning pipeline to tease out the most important factors in the long-term stability of intracortical recordings, including the electrode-design variables described above.
> >
> > Typo on the methods section:
> >
> > Thank you for catching this typo, we have corrected this issue.
> >
> > Ethics concerns:
> >
> > We have updated the manuscript with a section for the Ethics Statement. This statement indicates the use of animal studies and the corresponding approval by the Institutional Animal Care and Use Committee.
> >
> > Empirical novelty and significance/Aspects of the contributions exist in prior work:
> >
> > We consider that this study has important empirical novelty and is substantially different from previous work for the following reasons: 1) This is a unique dataset coming from a custom-made device implanted longitudinally in animals. 2) To our knowledge, this is the first longitudinal BCI study applying ML to predict whether a channel will be active. 3) Additionally, no ML explainability tools have been applied on similar datasets to elucidate the most important factors involved in the prediction of active channels over time.

---

### Decision · Program_Chairs · 2023-01-20

**Decision:**

Reject

**Justification For Why Not Higher Score:**

The technical contribution is limited and experimental evaluation is less convincing.

**Justification For Why Not Lower Score:**

N/A

**Metareview: Summary, Strengths And Weaknesses:**

Summary: This paper uses explainable machine learning methods to study which factors impact the number of active channels in a BCI. The authors analyze a number of important factors such a intersubject variability, time post-implantation, electrochemical properties and electrode design. Results are presented from an animal experiment.

Strengths: Well designed study with novel dataset that is highly relevant to the neuroscience/BCI/machine learning community.

Weakness: The contribution of this work is a bit over-claimed and it is not clear that the paper answers the question of when a channel will become inactive or assesses the longitudinal behaviour of channels over time. It is not clear what additional insight is revealed by the model. The experiments are less convincing on how predicting whether a channel is active, and consequently the proportion of active channels is related to the downstream BCI task.